# CLAA: Cross-Layer Attention Aggregation for Accelerating LLM Prefill

## Abstract

The prefill stage in long-context LLM inference remains a computational bottleneck. Recent token-ranking heuristics accelerate inference by selectively processing a subset of semantically relevant tokens. However, existing methods suffer from unstable token importance estimation, often varying significantly between layers. Evaluating token-ranking quality independently from heuristic-specific architectures is challenging. To address this, we introduce an Answer-Informed Oracle, which defines ground-truth token importance by aggregating attention from generated answers back to the prompt. Using this oracle, we find that existing heuristics (GemFilter and FastKV) exhibit substantial instability across layers, motivating our proposed heuristic of Cross-Layer Attention Aggregation (CLAA). CLAA robustly aggregates token scores across multiple consecutive layers, significantly improving ranking stability and achieving a new state-of-the-art performance. On LongBench, CLAA reduces Time-to-First-Token (TTFT) by up to 39% compared to the Full KV Cache baseline. At a similar level of task accuracy, CLAA provides this speedup while being over 10% faster than the prior state-of-the-art, FastKV, demonstrating a superior accuracy-speed tradeoff.

## 1 Introduction

Accelerating LLM inference on long-context prompts requires directly addressing the computational bottleneck of the initial prefill stage. This initial forward pass, which populates the KV cache for decoding, is compute-intensive due to both the quadratically-scaling self-attention mechanism and the large, linearly-scaling matrix multiplications of the MLP layers. While attention costs dominate at extreme lengths, the MLP computations are a significant factor for even moderately long prompts (e.g., 10k tokens), collectively contributing to a high Time-to-First-Token (TTFT)

One emerging framework to reduce this high computational cost is to operate on a smaller subset of only the most important prompt tokens. For instance, when processing a long document, a significant portion of the tokens can be irrelevant to the posed query. This insight leads to a common two-step framework: first, a computationally inexpensive heuristic is used to rank the importance of all prompt tokens. Then, the full and computationally expensive forward pass is performed only on this top-ranked subset, lowering the effective sequence length and directly reducing both attention and MLP costs.

The specific heuristics employed within this framework vary significantly in both their ranking signals and their architectural designs. For ranking, importance scores are derived from different sources. Methods like GemFilter Shi et al. (2024) and FastKV Jo et al. (2025) derive their ranking signal from the final tokens of the input prompt. In contrast, Speculative Prefill Liu et al. (2025) uses a separate smaller model to draft a plausible future continuation, which then serve as the basis for scoring the prompt. Architecturally, these approaches can differ significantly, with some restarting computation entirely, while others dynamically prune tokens during the forward pass. The variety of approaches makes principled comparison difficult. Current benchmarks rely on a single end-to-end performance metric, which conflates ranking quality with architectural efficiency and obscures how token importance evolves across different layers and attention heads.

To address these challenges, we propose an Answer-Informed Oracle (Figure 1) that establishes a ground-truth for evaluating token ranking heuristics. Our approach is founded on a key insight that the true importance of a prompt token is determined by the attention it receives from the generated

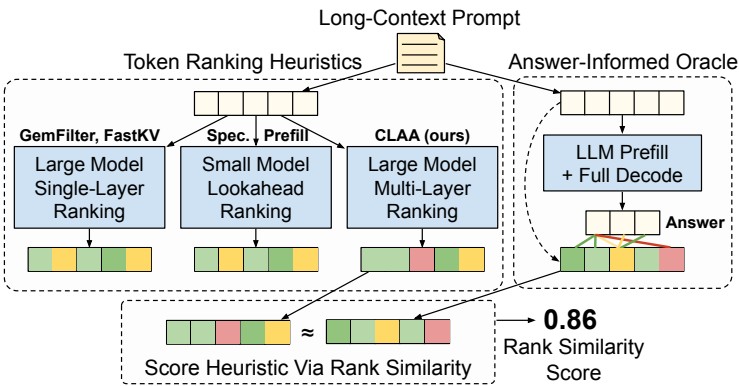

Figure 1: Illustration of our framework for evaluating token ranking heuristics for LLM Prefill acceleration. An Answer-Informed Oracle establishes a ground-truth token ranking by aggregating attention from the generated answer back to the prompt. This approach, which measures rank similarity between heuristic outputs and the oracle, motivates our Cross-Layer Attention Aggregation (CLAA) method that achieves higher alignment with the oracle.

answer. After a model produces a complete response, our oracle aggregates attention scores from the answer tokens back to each prompt token. This process creates a definitive ranking that enables principled evaluation via rank similarity metrics. Unlike end-to-end performance measures, our framework evaluates token importance separately from heuristic-specific architectural details.

Using this framework, we find that aggregating attention scores across multiple model layers produces a more robust signal of token importance. This insight leads to our proposal of Cross-Layer Attention Aggregation (CLAA), a simple and computationally efficient token ranking heuristic. By taking the maximum importance score for each token across multiple layers and heads, CLAA consistently produces rankings with higher similarity to our oracle, establishing a new state of the art on common long-context benchmarks like LongBench Bai et al. (2024) and RULER Hsieh et al. (2024). In summary, our main contributions are:

- We introduce an **Answer-Informed Oracle** that provides ground-truth for token importance by leveraging answer knowledge to identify critical tokens.

- We develop a **quantitative evaluation framework** using ranking similarity metrics for a principled comparison of token importance heuristics.

- We propose **Cross-Layer Attention Aggregation** (CLAA), a state-of-the-art token ranking heuristic derived from our systematic analysis.

## 2 RELATED WORK

### 2.1 APPROACHES TO PREFILL ACCELERATION

Hardware-aware optimizations such as FlashAttention Dao et al. (2022); Dao (2023) achieve substantial speedups through improved memory access patterns, though they still maintain quadratic complexity with respect to sequence length. Researchers have developed dynamic sparse attention methods to address this fundamental scaling limitation. Methods like SpAtten Wang et al. (2021), H2O Zhang et al. (2023), and MInference Jiang et al. (2024) leverage the observation that attention patterns vary dynamically with input content, adaptively identifying important tokens at runtime rather than relying on static sparsity patterns.

Alternative architectural solutions offer different trade-offs. State space models like Mamba Gu & Dao (2023) achieve linear complexity but face challenges with precise token recall, which has led to the development of hybrid architectures such as Jamba Lieber et al. (2024). Semantic compression techniques including LLMLingua Jiang et al. (2023); Pan et al. (2024) employ smaller auxiliary models to rewrite lengthy prompts into more concise representations, achieving substan-

tial compression ratios but introducing computational overhead from the compression process itself. Architectures like YOCO Sun et al. (2024) require additional training or fine-tuning phases.

Token-ranking heuristics take a different approach. These methods are training-free and reduce computational costs by identifying and processing only the most relevant token subsets. They can be applied directly to existing models without modification. This paper presents an evaluation framework and improved heuristic for such approaches.

## 2.2 TOKEN RANKING STRATEGIES IN PREFILL ACCELERATION

Several recent methods accelerate the prefill stage by avoiding a full-sequence forward pass through all layers. These approaches share a common strategy of first ranking the importance of prompt tokens using an initial, less computationally expensive ranking heuristic. This ranking allows the model to perform its main, compute-intensive forward pass on only a top-ranked subset of tokens, thereby lowering the total prefill cost. While these heuristics all rely on attention scores to perform this ranking, their primary distinction lies in the choice of query vectors (Q) used to probe the key vectors (K) of the prompt. To clarify these differences, we formalize each approach below using a consistent notation.

Let $T_{\text{prompt}}$ be an input prompt of length $L$. For a given model $M$ and layer $l$, let $K_{\text{prompt}}^{(l,h)} \in \mathbb{R}^{L \times d_k}$ be the matrix of key vectors for all prompt tokens at head $h$, where $d_k$ is the head dimensionality. The goal of each strategy is to compute an importance score $S_i$ for each prompt token $i \in \{1, \ldots, L\}$.

**GemFilter.** GemFilter Shi et al. (2024) hypothesizes that the query from the *very last token* of the prompt, after being processed by some initial layers, is sufficient to identify relevant context. It runs the model $M$ for $r$ layers to produce the query vector for the last token, $\mathbf{q}_{\text{last}}^{(r)}$. The importance score for the $i$-th prompt token is then computed by summing the raw attention scores (pre-softmax) from this single query across all attention heads $h$:

$$S_i^{\text{GF}} = \sum_h \left[ \frac{\mathbf{q}_{\text{last}}^{(r,h)} (K_{\text{prompt}}^{(r,h)})^\top}{\sqrt{d_k}} \right]_i \tag{1}$$

**FastKV.** FastKV Jo et al. (2025) uses a small observation window, $\mathcal{W}$, consisting of the $W$ most recent prompt tokens as queries. This allows a collective assessment from multiple positions at the end of the context. These queries are selected from a specific Token-Selective Propagation (TSP) layer, denoted $l_{\text{TSP}}$. The importance score is derived by summing the post-softmax attention probabilities from each query in the window across all attention heads $h$:

$$S_i^{\text{FKV}} = \sum_{j \in \mathcal{W}} \sum_h \left[ \text{Softmax} \left( \frac{\mathbf{q}_j^{(l_{\text{TSP}},h)} (K_{\text{prompt}}^{(l_{\text{TSP}},h)})^\top}{\sqrt{d_k}} \right) \right]_i \tag{2}$$

**Speculative Prefill.** In contrast, Speculative Prefill Liu et al. (2025) uses a separate, smaller *speculator model*, $M_{\text{spec}}$, to look into the "future." It generates $k$ lookahead tokens and uses their corresponding query vectors, $\{\mathbf{q}_{\text{gen},j}\}_{j=1}^k$, to score the prompt. This assesses token importance based on what a model *would* look for while generating a plausible continuation. The final score is the mean of the maximum raw attention scores (pre-softmax) from each lookahead query, taken across all layers and heads:

$$S_i^{\text{SP}} = \frac{1}{k} \sum_{j=1}^k \left( \max_{l,h} \left[ \frac{\mathbf{q}_{\text{gen},j}^{(l,h)} (K_{\text{prompt, spec}}^{(l,h)})^\top}{\sqrt{d_k}} \right]_i \right) \tag{3}$$

where $K_{\text{prompt, spec}}^{(l,h)}$ are the prompt key vectors as computed by the speculator model $M_{\text{spec}}$.

## 2.3 KV CACHE MANAGEMENT IN PREFILL ACCELERATION

In addition to pruning tokens during prefill, ranking heuristics are also used to compress the KV cache for decode.

**Compression via Sequence Pruning.** For methods like GemFilter and Speculative Prefill, KV cache compression is a direct consequence of their architectural design. These heuristics first identify a single, shared subset of $L_{\text{pruned}}$ important tokens. They then execute a second, main forward pass using only this pruned sequence. The compression follows naturally: the KV cache is built exclusively for these $L_{\text{pruned}}$ tokens. For any given layer $l$, the resulting key cache matrix, $K^{(l)} \in \mathbb{R}^{L_{\text{pruned}} \times d_k}$, is therefore uniform across all attention heads.

**Layer-wise Cache Compression.** In contrast, FastKV employs a dual-strategy architecture that performs explicit compression of the KV cache at each layer. At each layer $l$ before the final pruning step, it computes an importance score, $\mathcal{I}$, for each key-value head group $g$:

$$\mathcal{I}_{i,g}^{\text{KV-FKV}} = \frac{1}{|\mathcal{H}_g|} \sum_{h \in \mathcal{H}_g} \sum_{j \in \mathcal{W}} \left[ \text{Softmax} \left( \frac{\mathbf{q}_j^{(l,h)} (K_{\text{prompt}}^{(l,g)})^\top}{\sqrt{d_k}} \right) \right]_i \tag{4}$$

Based on these scores, a compressed KV cache is stored independently for each head group. Critically, while the stored cache is compressed, the full, unpruned hidden states are propagated to the subsequent layer for computation. Separating computation from cache storage lets the model process the full context while still maintaining a memory-efficient cache.

## 3 ANSWER-INFORMED ORACLE FRAMEWORK

### 3.1 ORACLE CONSTRUCTION

Our oracle is based on a key insight that prompt token importance is definitively measured by attention received from the generated answer. Unlike existing heuristics predicting future importance from partial information, our oracle leverages full knowledge of the generation process to determine precisely which prompt tokens influenced the response.

Algorithm 1 details the three-stage process for constructing oracle rankings. First, we process the full prompt through the model to extract key vectors for all prompt tokens. Second, we generate a complete answer while capturing the query vectors from each generated token. This separation into stages allows us to compute the exact attention scores we need without incurring the prohibitive memory cost of storing the full attention matrices from every generation step. Finally, we compute attention scores between all answer queries and prompt keys, aggregating these scores to produce a definitive importance ranking for each prompt token.

The oracle design incorporates several key architectural choices. We adopt a maximum aggregation strategy across layers and attention heads, similar to that used in Speculative Prefill Liu et al. (2025), to capture the peak importance of each token. This approach recognizes that even a single strong attention connection can indicate critical information. However, whereas Speculative Prefill aggregates attention from a handful of speculated future tokens, our oracle averages these peak scores across the entire ground-truth generated answer, providing a comprehensive measure of each token's true importance. Finally, we apply 1D average pooling with a small kernel to reduce local variations and stabilize rankings.

To quantitatively assess how well a ranking generated by a heuristic aligns with the ground truth established by the oracle, we use the Spearman Rank Correlation ($\rho$). This metric provides a global assessment of ranking quality by comparing the ranks assigned to each token by the oracle and heuristic methods. Values range from -1 (perfect disagreement) to 1 (perfect agreement), offering a direct measure of the ordinal similarity between the heuristic and oracle rankings.

### 3.2 ORACLE AS AN UPPER-BOUND BENCHMARK

Beyond comparing rankings via similarity metrics, the Answer-Informed Oracle establishes an empirical performance upper bound for any given token keep rate. This involves first computing the oracle ranking for a prompt using Algorithm 1, then executing a separate forward pass using only the top-k% tokens identified by this ranking.

To ensure a fair upper bound, the oracle-guided forward pass must precisely mirror the architectural strategy of the evaluated heuristic. For methods like GemFilter and FastKV that prune at an inter-

**Algorithm 1** Answer-Informed Oracle Token Ranking

**Require:** Oracle Model $M$, prompt tokens $T_{\text{prompt}}$, max output $N_{\text{gen}}$
1: $S_{\text{prompt}} \leftarrow \text{MODELFORWARD}(M, T_{\text{prompt}})$
2: $K_{\text{prompt}} \leftarrow \text{GETKEYS}(S_{\text{prompt}})$         ▷ Extract keys from prompt
3: Initialize empty lists $Q_{\text{gen}}, T_{\text{gen}}$
4: **for** $i = 1$ to $N_{\text{gen}}$ **do**         ▷ Generate oracle answer
5:     $S \leftarrow \text{MODELFORWARD}(M, T_{\text{prompt}} \oplus T_{\text{gen}})$
6:     $Q_{\text{gen}}[i] \leftarrow \text{GETLASTQUERY}(S)$         ▷ Store query
7:     $t \leftarrow \text{argmax}(\text{GETLASTLOGIT}(S))$
8:     **if** $t$ is EOS **then break**
9:     **end if**
10:     $T_{\text{gen}}[i] \leftarrow t$         ▷ Store token
11: **end for**
12: $A \leftarrow \text{COMPUTEATTN}(Q_{\text{gen}}, K_{\text{prompt}})$         ▷ All attention scores
13: $S_{\text{oracle}} \leftarrow \text{POOL1D}(\text{MEANAGG}(\text{MAXAGG}(A)))$         ▷ Aggregate & denoise
14: **return** $S_{\text{oracle}}$

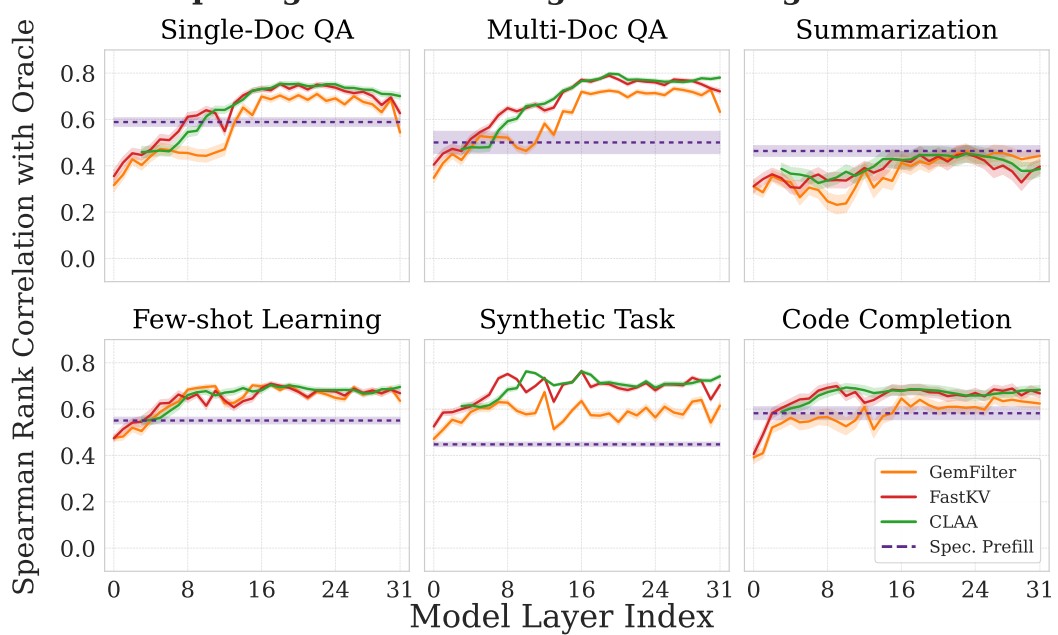

Figure 2: Layer-wise token ranking performance on Llama-3.1-8B-Instruct. Spearman correlation with answer-informed oracle across LongBench tasks, comparing existing heuristics to our proposed CLAA method.

mediate layer $l_p$, the oracle emulation follows a specific protocol: in layers before $l_p$, pre-computed oracle rankings compress the KV cache by selecting top-ranked token pairs, while full hidden states propagate forward to preserve the computational path. At the pruning layer $l_p$, these same rankings perform sequence pruning, filtering the hidden states themselves to retain only top-ranked tokens. This reduced sequence then flows through all subsequent layers, with the KV cache at $l_p$ compressed using the same indices.

# 4 CROSS-LAYER ATTENTION AGGREGATION (CLAA)

## 4.1 LIMITATIONS OF SINGLE-LAYER RANKING HEURISTICS

Figure 2 reveals two fundamental limitations of existing token-ranking methods when examined layer by layer against our oracle baseline. First, heuristics such as FastKV and GemFilter exhibit

substantial instability, with rank correlations fluctuating dramatically and experiencing sharp drops at specific layers. This volatility poses a significant risk for methods that depend on a single layer to determine sequence pruning decisions. When the selected layer (e.g., $l_{\text{TSP}}$) coincides with one of these performance troughs, the resulting token rankings become unreliable, potentially eliminating crucial tokens and compromising downstream task performance.

Second, our analysis reveals that early layers consistently show unreliable token rankings. The rank correlations in initial layers (particularly layers 0-4) are markedly lower than those in deeper layers. Basing pruning decisions on rankings from these unreliable early layers results in compression guided by low-fidelity signals. Consequently, essential key-value pairs may be prematurely discarded, undermining the model's ability to construct accurate semantic representations in subsequent layers and degrading the contextual information available for generation.

## 4.2 MULTI-LAYER AGGREGATION FOR ROBUST RANKING

Our analysis with the Answer-Informed Oracle reveals that single-layer heuristics can be unstable. To address this, we introduce Cross-Layer Attention Aggregation (CLAA), a simple yet efficient two-stage approach designed to improve stability in token importance assessment. First, given the unreliability of initial layers for ranking decisions, CLAA defers KV cache compression during the first $m = 4$ layers of the model. By maintaining the complete context in these early stages, we ensure that the model can construct high-fidelity semantic representations before any KV compression occurs. Second, rather than relying on potentially unstable single-layer rankings, CLAA aggregates token importance signals across multiple consecutive layers. By taking the maximum importance score across these layers, we ensure that tokens deemed critical by any layer are preserved, ensuring robustness against failures in layer-specific rankings. At a designated pruning layer $l_p$, CLAA synthesizes information from a window of $n$ preceding layers, $\mathcal{L} = \{l_p - n + 1, \ldots, l_p\}$. For each prompt token $i$ and layer $l' \in \mathcal{L}$, we compute a layer-specific importance score $S_i^{(l')}$ based on attention from a small observation window $\mathcal{W}$ containing the $W$ most recent prompt tokens:

$$S_i^{(l')} = \sum_{j \in \mathcal{W}} \sum_h \left[ \text{Softmax} \left( \frac{\mathbf{q}_j^{(l',h)} (K_{\text{prompt}}^{(l',h)})^\top}{\sqrt{d_k}} \right) \right]_i \tag{5}$$

Here, $\mathbf{q}_j^{(l',h)} \in \mathbb{R}^{d_k}$ represents the query vector from the $j$-th token in the observation window at layer $l'$ and head $h$, while $K_{\text{prompt}}^{(l',h)} \in \mathbb{R}^{L \times d_k}$ contains key vectors for all $L$ prompt tokens.

When the model reaches the pruning layer $l_p$, CLAA computes the final importance score for each token by taking the maximum across all collected layer scores: $S_i^{\text{CLAA}} = \max_{l' \in \mathcal{L}} S_i^{(l')}$. This maximum aggregation strategy preserves tokens deemed important by any recent layer while filtering layer-specific noise. Unlike single-layer methods, CLAA is resilient to the layer-specific variations shown in Figure 2. By synthesizing multiple perspectives, CLAA produces a stable ranking that more closely approximates the ground-truth importance identified by our Oracle, enabling effective KV cache compression.

## 5 EXPERIMENTS

### 5.1 EXPERIMENTAL SETUP

**Models and Datasets.** We evaluate our approach on Llama-3.2-3B-Instruct (28 layers) Grattafiori et al. (2024), Llama-3.1-8B-Instruct (32 layers), and Mistral-Nemo-12B-Instruct (40 layers) Mistral AI Team (2024). We evaluate token ranking quality using three benchmarks: (1) Long-Bench Bai et al. (2024) covers 16 English tasks across single/multi-document QA, summarization, few-shot learning, code completion, and synthetic tasks; (2) Needle-in-a-Haystack Kamradt (2023) tests information retrieval by embedding facts at various depths in 16K-64K token contexts; (3) RULER Hsieh et al. (2024) evaluates retrieval, multi-hop tracing, information aggregation, and QA across 12 subtasks using 64K contexts with 500 samples per subtask.

Table 3: RULER benchmark performance by category with 64K context on LLaMA-3.1-8B with 40% token keep rate.

| Method | Retr. | M-Hop | Agg. | QA | Avg. |
|--------|-------|-------|------|-----|------|
| FullKV | 98.27 | 86.88 | 86.67 | 63.30 | 83.78 |
| GemFilter | 82.23 | 58.20 | **88.33** | 63.70 | 73.12 |
| FastKV | 87.69 | 87.16 | 86.67 | 63.60 | 81.28 |
| CLAA | **89.85** | **87.72** | 86.67 | **63.80** | **82.01** |



Figure 3: Needle-in-a-Haystack result of LLaMA-3.1-8B-Instruct with 40% token keep rate. X denotes out of memory on 80GB A100.

**Baselines.** We compare CLAA against three recent token ranking heuristics. GemFilter Shi et al. (2024) uses attention from the last prompt token after processing through early layers. FastKV Jo et al. (2025) employs an observation window of recent tokens to score importance. Speculative Prefill Liu et al. (2025) uses Llama-3.2-1B-Instruct as a draft model to generate 8 lookahead tokens for scoring. For fair comparison across methods, we standardize several key parameters. To ensure an equitable comparison, a given token keep rate is applied consistently to both sequence pruning during prefill and KV cache compression for decoding across all methods. For both FastKV and CLAA, the observation window size ($W$) is set to 8 tokens. To stabilize token rankings, we apply 1D average pooling with a kernel size of 7 to the importance scores for all applicable methods. Unless otherwise specified, all layer-based ranking methods (GemFilter, FastKV, and CLAA) use layer 15 of the respective model as the pruning layer, a choice consistent with prior work Jo et al. (2025). For CLAA, we set the cross-layer aggregation window size ($n$) to 4, which is based on analysis in 5.4. Additionally, for CLAA, we keep the KV cache for the first $m = 4$ layers uncompressed.

**Evaluation Details.** For task evaluation, we report task-specific metrics from LongBench (F1 for QA tasks, Rouge-L for summarization, accuracy for classification) and binary retrieval accuracy for Needle-in-a-Haystack. For the RULER benchmark, we report the official recall-based accuracy averaged across its tasks. Experiments used a single A100 GPU (80GB). For the oracle, we use the same main model to generate complete answers. All settings use greedy decoding (temperature 0).

## 5.2 MAIN RESULTS

Our experiments reveal two key findings that validate CLAA's design principles. First, CLAA consistently outperforms existing methods across all compression rates. On LongBench with Llama-3.1-8B (Table 1), CLAA achieves 47.13% average performance at 10% token retention—within 0.7% of the answer-informed oracle (47.83%) and surpassing the next best method, FastKV, by 0.32%. This enables a more efficient accuracy-TTFT tradeoff, with CLAA at a 20% token keep rate achieving higher accuracy than FastKV at a 40% token keep rate (48.12% versus 47.68%). In general, the trend holds across model scales (Table 2), confirming that multi-layer aggregation provides more reliable token selection than single-layer approaches. Second, CLAA demonstrates greater robustness in challenging retrieval tasks. The Needle-in-a-Haystack results (Figure 3) reveal critical weaknesses in baseline methods. GemFilter entirely fails to retrieve needles at intermediate document positions (22-44% depth), whereas SpecPrefill consistently misses needles located in the latter half of documents. In contrast, CLAA maintains consistent retrieval accuracy across all needle positions, achieving the highest average score (0.909). In RULER (Table 3), where CLAA particularly excels at retrieval (89.85%) and multi-hop reasoning (87.72%), both tasks requiring the identification and integration of distributed information.

## 5.3 EFFICIENCY AND ACCURACY TRADE-OFF

Figure 4 shows that CLAA provides the most favorable accuracy-speed profile, closely tracking the performance of the Oracle. At a 10% keep rate, CLAA reduces TTFT by 39% compared to FullKV (from roughly 900ms to 550ms). It achieves 47.13% accuracy, closely approaching the oracle at 47.83%. FastKV offers the next best performance. In contrast, GemFilter and Speculative Prefill exhibit a larger accuracy degradation for a similar reduction in TTFT. Figure 5 provides a detailed breakdown of prefill and decode performance. A key design choice in CLAA is leaving the

Table 1: LLaMA-3.1-8B LongBench results across varying token keep rate (%).

| Method | Single-Document QA | | | Multi-Document QA | | | Summarization | | | Few-shot Learning | | | Synthetic | | Code | | |
| | NrtvQA | Qasper | MF-en | HotpotQA | 2WikiMQA | MuSiQue | GovReport | QMSum | MultiNews | TREC | TriviaQA | SAMSum | PCount | PRe | LCC | RB-P | Avg. |
|---|---|---|---|---|---|---|---|---|---|---|---|---|---|---|---|---|---|
| **Keep Token Rate = 100%** | | | | | | | | | | | | | | | | | |
| FullKV | 30.16 | 45.53 | 54.94 | 56.02 | 46.66 | 31.28 | 35.12 | 25.28 | 27.25 | 73.00 | 91.65 | 43.80 | 8.88 | 99.50 | 63.38 | 56.73 | **49.32** |
| **Keep Token Rate = 10%** | | | | | | | | | | | | | | | | | |
| Oracle | 29.85 | 43.94 | 55.85 | 54.99 | 47.11 | 28.92 | 32.26 | 25.29 | 21.95 | 68.50 | 91.43 | 41.80 | 7.47 | 99.50 | 59.63 | 56.85 | 47.83 |
| GemFilter | 24.36 | 21.07 | 39.73 | 51.29 | 33.92 | 25.78 | 28.94 | 18.98 | 17.42 | 60.50 | 91.53 | 40.39 | 4.76 | 87.50 | 22.73 | 32.47 | 37.59 |
| FastKV | 30.60 | 38.96 | 53.61 | 54.87 | 44.73 | 30.09 | 28.08 | 24.57 | 20.93 | 70.00 | 92.38 | 42.69 | 6.56 | 99.00 | 58.43 | 53.49 | 46.81 |
| SpecPrefill | 28.53 | 32.86 | 51.94 | 54.33 | 40.80 | 29.66 | 27.47 | 22.43 | 19.76 | 62.50 | 89.31 | 40.14 | 4.40 | 66.08 | 50.49 | 51.09 | 41.99 |
| CLAA | 31.09 | 42.36 | 53.68 | 53.83 | 44.73 | 31.53 | 28.15 | 24.76 | 20.42 | 70.00 | 92.37 | 42.93 | 6.51 | 99.50 | 58.31 | 53.86 | **47.13** |
| **Keep Token Rate = 20%** | | | | | | | | | | | | | | | | | |
| Oracle | 29.86 | 44.37 | 55.33 | 54.57 | 47.17 | 29.83 | 33.96 | 24.92 | 24.39 | 69.50 | 91.63 | 42.93 | 5.41 | 99.50 | 63.08 | 57.74 | 48.39 |
| GemFilter | 25.86 | 30.74 | 47.30 | 57.95 | 43.72 | 29.64 | 31.19 | 20.92 | 21.06 | 64.00 | 92.75 | 40.92 | 6.18 | 97.00 | 29.17 | 38.22 | 42.29 |
| FastKV | 30.54 | 40.45 | 54.42 | 53.67 | 47.25 | 30.12 | 28.49 | 24.33 | 21.46 | 70.50 | 92.08 | 42.85 | 7.09 | 99.50 | 59.68 | 54.93 | 47.33 |
| SpecPrefill | 27.41 | 40.09 | 56.67 | 56.39 | 40.05 | 27.55 | 30.28 | 23.61 | 22.52 | 62.50 | 91.04 | 41.19 | 3.00 | 87.50 | 52.90 | 50.49 | 44.57 |
| CLAA | 30.63 | 43.95 | 53.85 | 54.61 | 46.09 | 31.01 | 30.72 | 25.33 | 23.14 | 71.00 | 92.08 | 43.14 | 7.61 | 99.50 | 61.21 | 56.07 | **48.12** |
| **Keep Token Rate = 40%** | | | | | | | | | | | | | | | | | |
| Oracle | 30.14 | 45.74 | 55.36 | 54.27 | 47.42 | 30.95 | 35.11 | 25.17 | 25.97 | 70.00 | 91.79 | 42.56 | 6.84 | 99.50 | 63.99 | 57.33 | 48.88 |
| GemFilter | 26.90 | 39.14 | 52.33 | 56.02 | 46.85 | 32.22 | 33.56 | 22.80 | 24.36 | 68.50 | 91.67 | 43.84 | 4.65 | 99.50 | 37.34 | 42.04 | 45.11 |
| FastKV | 30.39 | 40.92 | 55.09 | 55.40 | 47.36 | 31.02 | 28.21 | 24.39 | 22.02 | 71.00 | 91.87 | 42.89 | 6.09 | 99.50 | 61.32 | 55.47 | 47.68 |
| SpecPrefill | 27.05 | 44.00 | 54.74 | 56.27 | 44.05 | 30.65 | 32.67 | 24.93 | 24.88 | 67.50 | 91.30 | 42.98 | 1.89 | 97.00 | 55.90 | 49.00 | 46.55 |
| CLAA | 30.37 | 45.84 | 54.32 | 55.02 | 47.38 | 31.72 | 33.10 | 24.87 | 25.21 | 71.00 | 92.04 | 43.31 | 7.18 | 99.50 | 62.40 | 56.23 | **48.72** |

Table 2: Average LongBench results across different models and token keep rates.

| Method | Keep Rate = 10% | | | Keep Rate = 20% | | | Keep Rate = 40% | | |
| | LLaMa (3B) | LLaMa (8B) | Mistral (12B) | LLaMa (3B) | LLaMa (8B) | Mistral (12B) | LLaMa (3B) | LLaMa (8B) | Mistral (12B) |
|---|---|---|---|---|---|---|---|---|---|
| FullKV | 44.23 | 49.32 | 48.32 | 44.23 | 49.32 | 48.32 | 44.23 | 49.32 | 48.32 |
| Oracle | 44.13 | 47.83 | 47.75 | 44.35 | 48.39 | 48.27 | 44.52 | 48.88 | 48.24 |
| GemFilter | 35.20 | 37.59 | 36.60 | 40.93 | 42.29 | 40.89 | 41.37 | 45.11 | 44.83 |
| FastKV | 42.45 | 46.81 | 45.62 | 42.45 | 47.33 | **47.38** | **44.17** | 47.68 | **48.04** |
| SpecPrefill | 34.06 | 41.99 | — | 35.90 | 44.57 | — | 39.17 | 46.55 | — |
| CLAA | **42.93** | **47.13** | **46.05** | **43.78** | **48.12** | 46.99 | 44.15 | **48.72** | 48.02 |

first four layers uncompressed to maintain foundational token representations before aggregation. This results in a minimal increase in KV cache size, for example, 0.3 GB for CLAA versus 0.1 GB for the Oracle at a 10% keep rate. This modest increase in memory is justified by significant accuracy improvements (Figure 4), highlighting a favorable trade-off between resource usage and performance. In contrast, the GemFilter approach results in a significantly larger KV cache (1.3 GB at a 10% keep rate) because its implementation retains the full cache and indexes into it during decoding, which negatively impacts decode throughput (16 tps compared to 19-20 tps for others).

## 5.4 ABLATION STUDIES

**Impact of Pruning Layer.** Figure 6 (left) demonstrates a strong positive correlation between LongBench accuracy and the pruning layer index. For all heuristics, applying pruning at later layers results in higher accuracy. For instance, the accuracy of FastKV increases from 32.5% at layer 3 to 46.5% at layer 15. This finding indicates that premature pruning at early layers, which compute foundational representations, leads to significant information loss. CLAA consistently outperforms both FastKV and GemFilter across all tested layers, particularly at earlier ones, which underscores the robustness of cross-layer aggregation. As the pruning layer index increases, the performance of

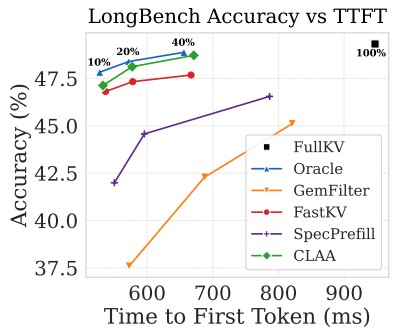

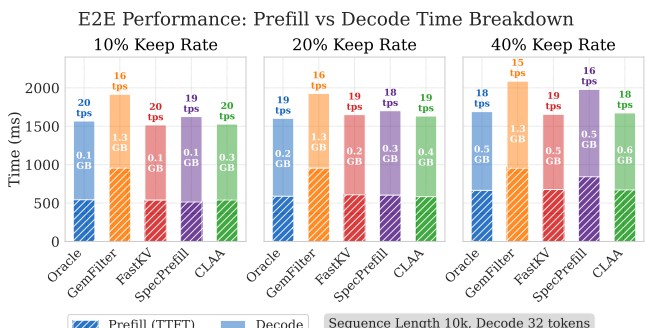

Figure 4: LongBench accuracy versus Time-to-First-Token (TTFT) for LLaMA-3.1-8B-Instruct on a 10k token sequence. Points correspond to 10%, 20%, and 40% keep rates.

Figure 5: End-to-end performance breakdown for a 10k token prompt and 32 token generation. Bars show Prefill (TTFT) and Decode time. Annotations indicate decode throughput (tokens per second) and KV cache size (GB) at the start of decode.

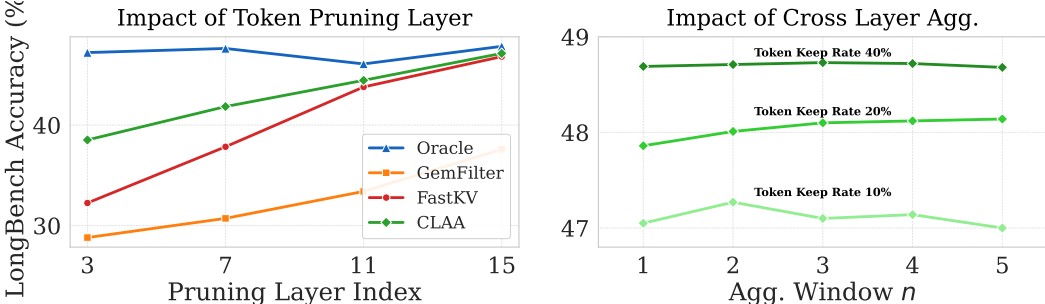

Figure 6: **Left**: Impact of the pruning layer index on LongBench accuracy. Pruning later in the model improves performance for all heuristics. **Right**: Impact of the cross-layer aggregation window size ($n$) at different token keep rates. Increasing the window size improves accuracy.

all methods improves and converges toward the upper bound established by the oracle. The selection of layer 15 for our main experiments, positioned approximately halfway through the model, is therefore justified as a balance between computational savings and performance.

**Impact of Cross-Layer Aggregation Window.** Figure 6 (right) illustrates the effect of the aggregation window size, $n$, on the performance of CLAA at different token keep rates. For all keep rates, expanding the window from $n = 1$ to $n = 2$ provides a substantial accuracy boost, validating the core hypothesis that cross-layer aggregation mitigates the instability of single-layer ranking. Larger windows have nuanced effects, with performance for higher token keep rates (20% and 40%) typically stabilizes around $n = 4$. For the highly aggressive 10% keep rate, performance is more volatile, peaking at $n = 2$. These results demonstrate that a small aggregation window is sufficient to stabilize rankings. A window size of $n = 4$ was selected for our experiments as it offers a robust performance benefit across various keep rates with minimal overhead.

## 6 CONCLUSION

We introduce the Answer-Informed Oracle, a novel framework that establishes ground-truth for token importance in evaluating prefill acceleration heuristics. Through systematic analysis with this oracle, we uncover that existing methods suffer from significant instability in token rankings across model layers. To address this limitation, we propose Cross-Layer Attention Aggregation (CLAA), which stabilizes rankings by aggregating importance scores across consecutive layers. CLAA achieves state-of-the-art performance, improving accuracy over previous methods while reducing Time-to-First-Token by up to 39%.

## ETHICS STATEMENT

Our work focuses on the computational efficiency of Large Language Models. All experiments were conducted using publicly available, pre-trained models and established academic benchmarks. This study did not involve human subjects or the use of personally identifiable information.

We recognize that technologies improving LLM efficiency have a dual-use nature, as they can reduce computational costs for both beneficial and malicious applications. Furthermore, the act of pruning input tokens based on attention is not neutral. It can inadvertently alter the behavior of a model, potentially skewing degrading performance in subtle ways. We recommend future work to characterize the impact of such token-pruning methods on model robustness across a wider range of tasks and languages.

## REPRODUCIBILITY STATEMENT

All experiments were conducted using publicly available models and benchmarks. To ensure our results can be independently verified, we provide two key resources in the appendix. Appendix A contains a comprehensive table (Table 4) detailing all hyperparameter configurations for our proposed method and for every baseline, including values explored during our ablation studies. Appendix D provides unified, line-by-line pseudocode for all methods evaluated (Listings 1–5). We commit to releasing a public, open-source implementation of our work upon publication. A link to the repository will be provided in the camera-ready version of this paper.

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

## A  HYPERPARAMETER CONFIGURATION

This appendix details the hyperparameter configurations for all methods to ensure a transparent and reproducible comparison. Key architectural parameters, such as the pruning layer index, were swept across all applicable methods for our ablation studies (Section 5.4). The final values used in our main results (e.g., Tables 1 and 2) were selected to provide a consistent and fair comparison point across methods.

## B  ABLATION ON FIRST COMPRESSION LAYER ($m$)

In our main experiments, we set the number of initial uncompressed layers for CLAA to $m = 4$. This decision was based on an ablation study investigating the trade-off between early-layer compression and model accuracy. As shown in Figure 7, deferring compression to later layers consistently improves performance across all token keep rates. Starting compression from layer 0 (i.e., $m = 0$) results in the lowest accuracy, as it forces the model to build its initial representations from a compressed context. Delaying compression until layer 4 provides a significant accuracy boost, justifying the choice of $m = 4$ as a default that balances performance and computational efficiency.

## C  PER-TASK RANKING CORRELATION ANALYSIS

This section analyzes token ranking performance by comparing various heuristics against the Answer-Informed Oracle. Figure 8 shows Spearman Rank Correlation for each heuristic across all model layers.

Table 4: Hyperparameter settings for all methods evaluated on LLaMA-3.1-8B-Instruct.

| Method | Parameter | Symbol | Value(s) Explored | Final Value in Main Results |
|---|---|---|---|---|
| **Oracle** (Ours) | Token Keep Rate | - | {0.1, 0.2, 0.4} | Main Variable |
| | Pruning Layer Index | $l_p$ | {3, 7, 11, 15, 19} | 15 |
| **FastKV** | Token Keep Rate | - | {0.1, 0.2, 0.4} | Main Variable |
| | TSP Layer Index | $l_{\text{TSP}}$ | {3, 7, 11, 15, 19} | 15 |
| | Observation Window | $W$ | {8} | 8 |
| | Pooling Kernel Size | - | {7} | 7 |
| **GemFilter** | Token Keep Rate | - | {0.1, 0.2, 0.4} | Main Variable |
| | Routing Layer Index | $r$ | {3, 7, 11, 15, 19} | 15 |
| **SpecPrefill** | Token Keep Rate | - | {0.1, 0.2, 0.4} | Main Variable |
| | Lookahead Tokens | $k$ | {8} | 8 |
| **CLAA** (Ours) | Token Keep Rate | - | {0.1, 0.2, 0.4} | Main Variable |
| | Pruning Layer Index | $l_p$ | {3, 7, 11, 15, 19} | 15 |
| | Aggregation Window | $n$ | {1, 2, 4, 8, 12} | 4 |
| | First Uncompressed Layer | $m$ | {0, 2, 4, 6} | 4 |
| | Observation Window | $W$ | {8} | 8 |
| | Pooling Kernel Size | - | {7} | 7 |

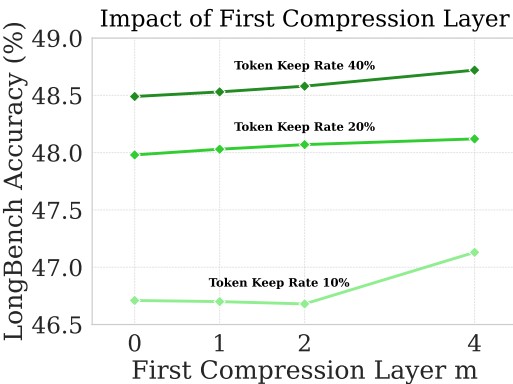

Figure 7: Impact of the first compression layer index ($m$) on LongBench accuracy for CLAA. For $m > 0$, layers 0 to $m - 1$ are kept uncompressed. The experiment shows that deferring compression to later layers (increasing $m$) improves accuracy, especially for higher token keep rates. These results are on Llama-3.1-8B-Instruct.

- **Layer-specific volatility:** Single-layer heuristics show significant performance drops at specific layers. In NarrativeQA, GemFilter and FastKV exhibit sharp correlation drops around layer 10. This pattern repeats across QASPER, HotpotQA, and QMSum. CLAA avoids these drops through multi-layer aggregation.

- **Practical implications:** Methods using single pre-selected layers risk severe performance degradation if their chosen layer coincides with a performance trough. CLAA provides more reliable token selection by aggregating across a layer window.

- **Performance convergence:** While layer-dependent heuristics reach similar high performance in deep layers (20-31), CLAA achieves comparable results without the intermediate volatility.

- **Universal patterns:** Early layers (0-8) consistently fail to predict token importance across all methods. Speculative Prefill serves as a task-dependent baseline with varying competitiveness.

These results demonstrate that CLAA provides more stable and reliable token ranking compared to single-layer methods, making it a robust choice for prefill acceleration.

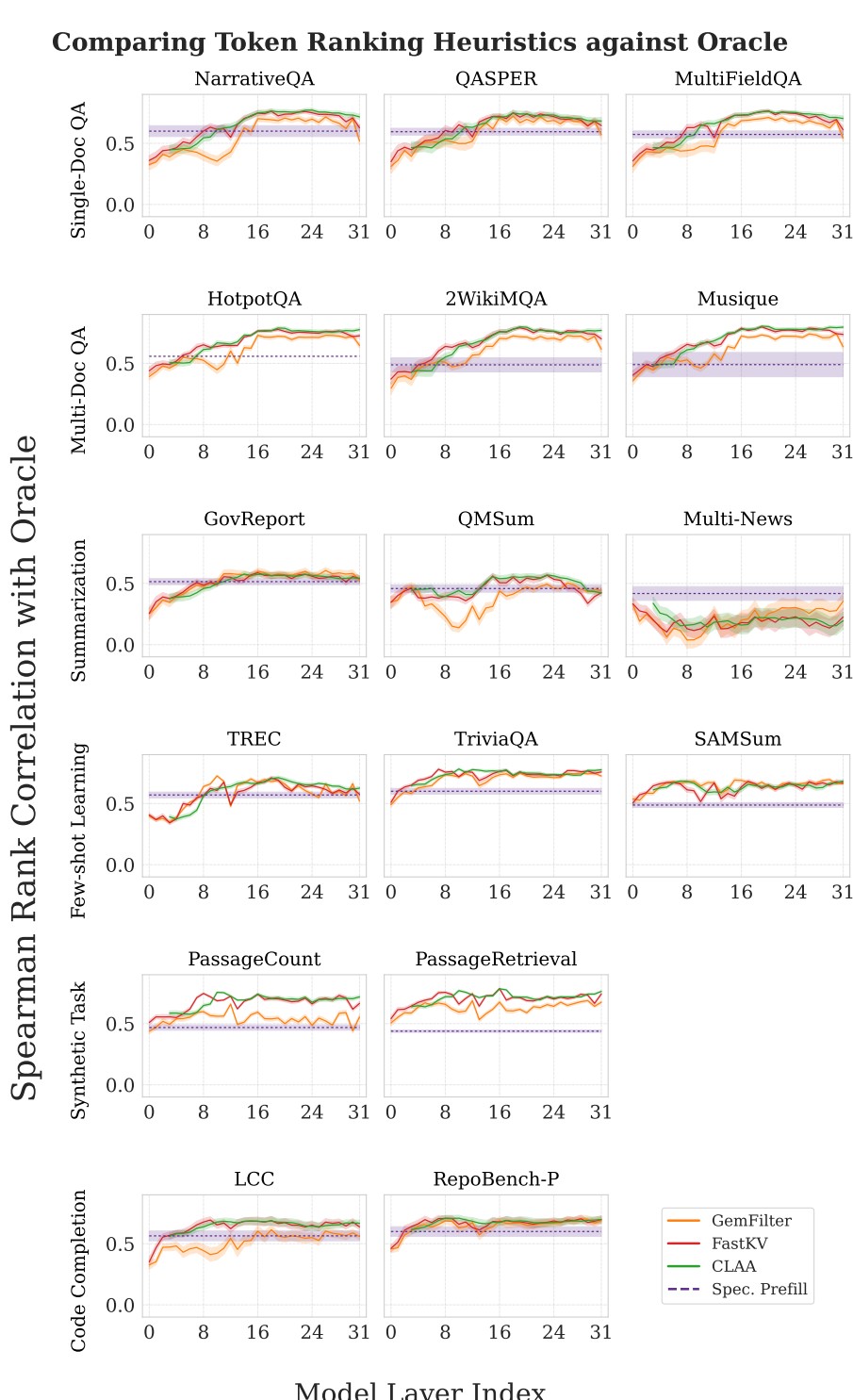

Figure 8: Detailed per-task comparison of token ranking heuristics against the Answer-Informed Oracle.

# D   TOKEN RANKING HEURISTIC PSEUDOCODE

This section provides detailed and unified pseudocode for the token ranking heuristics discussed in the paper: GemFilter, FastKV, Speculative Prefill, our Answer-Informed Oracle, and our proposed method, CLAA.

## D.1   GEMFILTER

GemFilter operates in a two-pass system. It first runs a partial forward pass to rank tokens using the query from the final prompt token. It then discards this intermediate state and executes a new, standard forward pass using only the top-ranked tokens and their original positions. This architectural approach is outlined in Listing 1.

## D.2   FASTKV

Unlike GemFilter's two-pass system, FastKV operates in a single, continuous forward pass. At each layer, it uses an "observation window" of recent tokens to rank all prompt tokens and compress the stored KV cache. At a designated Token-Selective Propagation (TSP) layer, it performs a one-time sequence pruning, after which the forward pass continues on the reduced sequence. This integrated architecture is outlined in Listing 2.

## D.3   SPECULATIVE PREFILL

Speculative Prefill uses a two-model architecture. A small "speculator" model first performs a full prefill and generates a few lookahead tokens. The queries from these generated tokens are used to rank the importance of the original prompt tokens. Finally, the large "base" model performs a selective prefill on only the top-ranked prompt tokens, using their original position IDs to maintain context. This multi-stage process is outlined in Listing 3.

## D.4   ANSWER-INFORMED ORACLE

The Answer-Informed Oracle provides a ground-truth token ranking by leveraging knowledge of the final generated answer. Its operation is split into two phases. First, in an offline ranking phase, it generates a complete answer and measures the attention from each generated token back to the original prompt tokens. Second, in an online evaluation phase, this pre-computed ranking is used to perform a selective prefill, establishing a theoretical performance upper bound. This process is detailed in Listing 4.

## D.5   CROSS-LAYER ATTENTION AGGREGATION (CLAA)

Our proposed method, CLAA, enhances the single-pass architecture by improving ranking stability. It first defers any compression for an initial $m$ layers. Then, for each subsequent layer, it computes importance scores using an observation window and stores them in a buffer of size $n$. At a designated pruning layer, $l_p$, it aggregates scores across this buffer by taking the maximum value for each token. This process is outlined in Listing 5.

```python
def gemfilter_prefill(model, prompt_tokens, routing_layer, keep_rate):
    # Run a partial forward pass to get the query from the last prompt
    token
    hidden_states = model.forward(prompt_tokens,
    stop_at_layer=routing_layer)
    q_last = hidden_states.get_last_token_query(layer=routing_layer)
    k_all = hidden_states.get_keys(layer=routing_layer)

    # Rank tokens by their raw attention (pre-softmax) to the last token
    q
    scores = aggregate_attention(q_last, k_all, use_softmax=False)
    top_k_indices = topk(scores, keep_rate).indices

    pruned_tokens = gather(prompt_tokens, top_k_indices)

    # Restore position ids
    original_position_ids = gather(range(len(prompt_tokens)),
    top_k_indices)

    # Execute a full forward pass on pruned sequence
    final_outputs, final_kv_cache = model.forward(
        pruned_tokens,
        position_ids=original_position_ids
    )

    return final_outputs, final_kv_cache
```

Listing 1: GemFilter Prefill Logic.

```python
def fastkv_prefill(model, prompt_tokens, tsp_layer, window_size,
    keep_rate):
    hidden_state = model.embed(prompt_tokens)
    kv_cache = {}

    for l in range(model.num_layers):
        # Get queries from the observation window
        q_window = hidden_state.get_last_n_queries(n=window_size,
    layer=l)
        k_all = hidden_state.get_keys(layer=l)
        v_all = hidden_state.get_values(layer=l)

        # Rank tokens using post-softmax attention from the window
        scores = aggregate_attention(q_window, k_all, use_softmax=True)
        top_k_indices = topk(scores, keep_rate).indices

        # Compress KV cache using rankings (for decode)
        kv_cache[l] = gather(k_all, v_all, on_indices=top_k_indices)

        hidden_state = model.layer_forward(l, hidden_state,
    use_kv=(k_all, v_all))

        # At the TSP layer, prune hidden state with rankings
        if l == tsp_layer:
            hidden_state = gather(hidden_state, on_indices=top_k_indices)

    final_outputs = model.final_norm(hidden_state)
    return final_outputs, kv_cache
```

Listing 2: FastKV Prefill Logic.

```
def speculative_prefill(base_model, spec_model, prompt_tokens,
    look_ahead_k, keep_rate):
    # Run a small speculator model on the full prompt
    spec_outputs, spec_kv_cache = spec_model.forward(prompt_tokens)
    spec_k_prompt = spec_kv_cache.get_all_keys()

    # Generate 'look_ahead_k' tokens with the speculator to get qs
    q_generated = []
    next_token = spec_outputs.get_next_token()
    for _ in range(look_ahead_k):
        lookahead_out, spec_kv_cache = spec_model.forward(next_token,
    spec_kv_cache)
        q_generated.append(lookahead_out.get_query())
        next_token = lookahead_out.get_next_token()

    # Rank prompt tokens using draft qs
    scores = aggregate_attention(q_generated, spec_k_prompt)
    top_k_indices = topk(scores, keep_rate).indices

    pruned_tokens = gather(prompt_tokens, top_k_indices)
    original_position_ids = gather(range(len(prompt_tokens)),
    top_k_indices)
    final_outputs, final_kv_cache = base_model.forward(
        pruned_tokens,
        position_ids=original_position_ids
    )
    return final_outputs, final_kv_cache
```

Listing 3: Speculative Prefill Logic.

```
def get_oracle_ranking(model, prompt_tokens, max_gen_len):
    _, prefill_kv_cache = model.forward(prompt_tokens)
    k_prompt = prefill_kv_cache.get_all_keys()

    q_generated = []
    next_token = prefill_kv_cache.get_next_token()
    current_kv_cache = prefill_kv_cache
    while not is_eos(next_token) and len(q_generated) < max_gen_len:
        output, current_kv_cache = model.forward(next_token,
    past_kv=current_kv_cache)
        q_generated.append(output.get_query())
        next_token = output.get_next_token()

    scores = aggregate_attention(q_generated, k_prompt,
    use_softmax=False)

    return scores

def oracle_prefill(model, prompt_tokens, oracle_scores, keep_rate):
    # Use the pre-computed oracle scores to select the top-k indices
    top_k_indices = topk(oracle_scores, keep_rate).indices

    pruned_tokens = gather(prompt_tokens, top_k_indices)
    original_position_ids = gather(range(len(prompt_tokens)),
    top_k_indices)

    final_outputs, final_kv_cache = model.forward(
        pruned_tokens,
        position_ids=original_position_ids
    )
    return final_outputs, final_kv_cache
```

Listing 4: Answer-Informed Oracle Prefill Logic.

```python
def claa_prefill(model, prompt_tokens, pruning_layer, aggregation_window,
                 defer_layers, window_size, keep_rate):
    hidden_state = model.embed(prompt_tokens)
    kv_cache = {}
    layer_scores_buffer = collections.deque(maxlen=aggregation_window)

    for l in range(model.num_layers):
        # Defer any compression for the first 'defer_layers'
        if l < defer_layers:
            hidden_state = model.layer_forward(l, hidden_state)
            kv_cache[l] = hidden_state.get_kv_pair()
            continue

        # Get queries from the observation window
        q_window = hidden_state.get_last_n_queries(n=window_size,
    layer=l)
        k_all, v_all = hidden_state.get_keys(layer=l),
    hidden_state.get_values(layer=l)

        # Compute importance scores for the current layer using the
    observation window
        current_layer_scores = aggregate_attention(q_window, k_all,
    use_softmax=True)
        # Store the scores in a rolling buffer for later aggregation.
        layer_scores_buffer.append(current_layer_scores)

        # Compress KV cache using rankings (for decode)
        compression_indices = topk(current_layer_scores,
    keep_rate).indices
        kv_cache[l] = gather(k_all, v_all,
    on_indices=compression_indices)

        hidden_state = model.layer_forward(l, hidden_state,
    use_kv=(k_all, v_all))

        # At the TSP layer, prune hidden state with agg. rankings
        if l == pruning_layer:
            # Aggregate scores (Eq. 6 in paper)
            aggregated_scores = max(layer_scores_buffer, dim=0)
            pruning_indices = topk(aggregated_scores, keep_rate).indices
            hidden_state = gather(hidden_state,
    on_indices=pruning_indices)

    final_outputs = model.final_norm(hidden_state)
    return final_outputs, kv_cache
```

Listing 5: Cross-Layer Attention Aggregation (CLAA) Prefill Logic.

