# OpenReview forum: "CLAA: Cross-Layer Attention Aggregation for Accelerating LLM Prefill"
_ICLR.cc/2026/Conference — Submitted to ICLR 2026_

### Official Review · Reviewer_a9cR · 2025-10-24

**Soundness:** 3
**Presentation:** 4
**Contribution:** 3
**Rating:** 8
**Confidence:** 4

**Summary:**

This paper tackles the prefill bottleneck in long-context LLM inference by proposing a novel token ranking heuristic called Cross-Layer Attention Aggregation (CLAA). The authors introduce an Answer-Informed Oracle that establishes ground-truth token importance by aggregating attention from generated answers back to the prompt. Using this oracle, they demonstrate that existing methods like GemFilter and FastKV suffer from unstable rankings across layers. CLAA addresses this by aggregating token scores across multiple consecutive layers, achieving up to 39% reduction in TTFT while maintaining accuracy. The method is evaluated on LongBench, Needle-in-a-Haystack, and RULER benchmarks using Llama-3.2-3B, Llama-3.1-8B, and Mistral-Nemo-12B models.

**Strengths:**

- Novel and principled evaluation framework: The Answer-Informed Oracle is a clever contribution that provides a ground-truth way to evaluate token ranking quality independent of architectural differences. The idea of using attention from the actual generated answer to score prompt tokens is intuitive and well-motivated. This makes it much easier to compare different heuristics fairly, which has been a real problem in this area.
- Strong empirical results with comprehensive evaluation: The experiments are pretty thorough - covering multiple benchmarks (LongBench, Needle-in-a-Haystack, RULER) and showing consistent improvements. The accuracy-speed tradeoff in Figure 4 clearly shows CLAA tracking the oracle much more closely than other methods.
- Well-motivated method design: The layer-wise analysis in Figure 2 really drives home why cross-layer aggregation makes sense. Seeing those sharp correlation drops at specific layers for single-layer methods makes the instability problem very concrete. The design choices of deferring compression for the first 4 layers and max aggregation across layers follow naturally from this analysis.
- Good ablation studies: The ablations on the pruning layer (Figure 6 left) and aggregation window size (Figure 6 right) help understand when and why CLAA works.
- Paper writing is good and reading-friendly.

**Weaknesses:**

- Limited model coverage in experiments: The paper only evaluates on Llama-3.2-3B, Llama-3.1-8B, and Mistral-Nemo-12B. It would be more convincing to see results on other model families like Qwen3 or larger models (e.g., 30B+ scale). Different architectures might have different attention patterns, and larger models might show different layer-wise stability characteristics. The current evaluation leaves some doubt about whether these findings generalize broadly across the model landscape.
- Missing discussion on failure cases and limitations: The paper doesn't really dig into when CLAA might struggle or perform poorly. Are there specific task types or prompt characteristics where cross-layer aggregation doesn't help? The Multi-News summarization results in Figure 2 show relatively flat correlations for all methods - what's happening there? Understanding failure modes would make the contribution more complete.

**Questions:**

NA. Overall, I like the paper.

---

> ### Author Response · Authors · 2025-11-21
>
> Thank you for your feedback identifying the value of the Answer-Informed Oracle Framework. We appreciate your constructive feedback in regards to model coverage and failure analysis.
>
> **Model Coverage**
>
> We agree that evaluating on larger parameter scales (e.g., 30B+) and different model families would strengthen the generalization of our findings. We plan to extend our evaluation to include larger models and newer ones (such as Qwen3-series) in the camera-ready version. There is a bit of engineering work to upgrade the transformers version currently used (4.45.0) to support these newer models.
>
> **Failure Cases and Multi-news Analysis**
>
> As you mention, there are several datasets (e.g., Multi-news) where all methods perform relatively flat in the rankings compared to the oracle. The heuristics estimate importance based on the state at the end of the prompt (or a few tokens into the future in the case of Speculative Prefill). In summarization tasks, we believe this importance is more dynamic and evolves throughout the generation (e.g., what tokens at the start of generation things are important may not agree with tokens later in the generation). This is less of an issue in retrieval tasks, where the question is typically directly related to the answer (i.e., finding the relevant snippets from the document). To us, this finding suggests that for tasks with evolving information needs, there is a theoretical ceiling for static prefill pruning. Future frameworks may need to re-assess token importance dynamically during the decoding phase to capture these shifts, rather than relying solely on a single snapshot at the prompt boundary. The challenge is how to do this effectively. We will add this discussion in a new Limitations section in the revised version.

---

### Official Review · Reviewer_Ke2f · 2025-10-28

**Soundness:** 3
**Presentation:** 3
**Contribution:** 2
**Rating:** 4
**Confidence:** 4

**Summary:**

The author first proposes a new way of defining token importance as an Oracle metrics to understand the quality of different token importance estimation techniques, whose results will be used to skip prefill computations and KV cache. Then the author introduces a method that aggregates attention scores across the first several layers, which they show to be more aligned with the Oracle they proposed. This method, like prior works, successfully reduces the TTFT while ensuring enough quality to downstream performance.

**Strengths:**

1. The author proposes a principle way of checking the validity of a token estimation method, upon which they design a new way of aggregating the attention mechanism to conduct token dropping.
2. The work compares many important baselines both from a quality and efficiency perspective.
3. LongBench, RULER, and Niah are used for long context evaluations, as well as both prefill TTFT and KV cache / memory profiling, which is very comprehensive.

**Weaknesses:**

1. Prior works have shown that token pruning can become less effective for shorter, standard tasks, which should be included for completeness.
2. The model focuses on the 8B model, and should ideally include other sizes to further prove its practicality.
3. The method seems more like an incremental approach from prior works that use attention as the main metrics for token estimation, the main novel part is how to aggregate the attention scores more robustly compared to GemFilter.
4. RULER results should be better with spec prefill.
5. For Speculative Prefill, the draft model needs to be large enough as tested in the original paper to have enough estimation capacity for preserving quality. Also, in Figure 5, the TPS as well as prefill TTFT seem too bad compared to what the paper mentions (probably due to not using their vllm implementation?) The reviewer recommends only mentioning CLAA’s superiority under the specific setting with 8B and non-original implementation for fairness.
6. Like all prior works, it will be great to mention if this method can handle multi-turn setting where token skipping methods could potentially fall short.
7. The method requires calculating q*K^T for each q in the prompt and averaged over the observational window, which is not natively outputted for SOTA attention kernels such as flash and flex attention. In order for this to be maximally performant, it will be better to mention this as a potential future work.
8. In Figure2, it is not obvious that CLAA is better than FastKV though.

**Questions:**

Listed above in weakness.

---

> ### Author Response · Authors · 2025-11-21
>
> Thank you for the detailed review and for highlighting both the strengths of the paper and the concerns around Speculative Prefill and FlashAttention.
>
> **Shorter-context Pruning**
>
> We agree that extending the analysis to shorter, standard tasks is a natural direction, and we will mention this explicitly as future work. However, we also want to mention that LongBench already provides a reasonable range of prompt lengths across tasks. For instance, using Llama-3.1-8B tokenizer, we have shorter average prompt lengths per datasets (e.g., MultiNews ~2.5k, LCC ~3k, Qasper ~5k) and longer ones (e.g., MuSiQue ~15k, NarrativaQA ~30k). We will add the average prompt length per dataset to Table 1 to make this more clear. We already see negative correlation between prompt length and performance across all pruning methods (to varying degrees).
>
> Once prompt length drops below ~2k, the benefit of token pruning is mostly mitigated. But, it is still useful to see how these token pruning methods break in such contexts.
>
> **Scope of evaluation**
>
> Table 2 includes results for 3B, 8B, and 12B models, and we will make this more prominent in the text. Our experiments focus on long-context workloads to align against comparisons made by prior work. We plan to extend our evaluation to include larger models 30B+ and newer ones (such as Qwen3-series) in the camera-ready version. There is a bit of engineering work to upgrade the transformers version currently used (4.45.0) to support these newer models.
>
> **On the incremental nature of CLAA**
>
> We agree that CLAA is an evolution of attention-based ranking methods (GemFilter, FastKV) rather than a paradigm shift. We view this simplicity as a potential strength for easier deployment. In Figure 2, we observe that there is some ranking instability for GemFilter/FastKV, relative to our Answer-Informed Oracle, around the typical pruning layers (e.g., layer 15) and therefore propose aggregation around those points to smooth out such noisy rankings.
>
> Our primary contribution is the oracle, which provides a ground-truth ranking and cleanly separates ranking quality from architectural details. We hope this tool can be used by the community to further improve token ranking heuristics beyond the layerwise aggregation approach we propose here.
>
>
> **Speculative Prefill Implementation Details**
>
> Our goal was to compare token-ranking heuristics under as controlled a setting as possible. In the current draft we did not explain this clearly enough.
>
> All methods in our experiments (e.g., GemFilter, FastKV, CLAA, and our Speculative Prefill baseline) run on the same Hugging Face stack with FlashAttention-2. We follow the design used in the public FastKV code: we subclass the attention module and insert a small scoring step after the standard `_flash_attention_forward` kernel. As a result, every method in our comparison uses FlashAttention-2 (including baseline FullKV).
>
> For Speculative Prefill in particular, we implemented the ranking heuristic so that, for a given prompt and 1B/8B model pair, the token rankings match those produced by the original authors’ public vLLM implementation. We verified this agreement directly. The absolute TTFT/TPS numbers in our figures differ from those reported in the Speculative Prefill paper because our engine (HF + FlashAttention-2, without PagedAttention) is different from their vLLM-based system. We will revise Section 5.1 to make this explicit and to state clearly that our Speculative Prefill results are intended for relative comparison within against other HF transformer implementations, not for a direct reuse of the original codebase.
>
> The RULER results should be read in this light. We intentionally use a 1B draft model to represent a resource-constrained deployment scenario, where running a much larger draft is not feasible. Under this constraint, Speculative Prefill underperforms the original paper’s configuration, but the setting is the same for all methods and the comparison remains fair. We will clarify that this design choice reflects a particular deployment regime rather than a change to the Speculative Prefill algorithm.
>
> It may also help to connect the Oracle more directly to Speculative Prefill. Conceptually, our Answer-Informed Oracle applies the same idea, that is measuring answer-conditioned importance of prompt tokens, but on the target model and across the full generated continuation. In that sense it can be viewed as a direct analogue of the Speculative Prefill perspective, specialized for evaluation rather than for runtime acceleration. We will add this connection to the revised version.

---

> > ### Author Response · Authors · 2025-11-21
> >
> > **Multi-turn Conversations**
> >
> > Our method does not address this concern and we will add a short section in related work to discuss this paradigm. All other token pruning methods run into the same issue (as they are estimating importance based on the first user question in some fashion). Theoretically, there is nothing preventing these methods from being run for each additional user turn in a long multi-conversation chain, except that at some point it would be more efficient to just compute full attention. We will leave this as future work.
> >
> > **Attention Scoring Overhead**
> >
> > In our implementation, all methods in the paper are already evaluated using Hugging Face’s FlashAttention-2 implementation. We adopt the same design pattern as the public FastKV implementation: we subclass the FlashAttention-2 attention module to intercept the attention forward pass to compute layer-wise importance scores $S^{(l)}$ (Eq. 5) using the subset of query vectors corresponding to the observation window, $Q_{\mathcal{W}}$, against the full key matrix $K_{\text{prompt}}$. The measured TTFT in Figures 4–5 is end-to-end wall-clock time on the FlashAttention-2 stack, already including this scoring and aggregation overhead. This $Q_{\mathcal{W}}$ by $K_{\text{prompt}}$ computation represents less than 2% overhead of the entire end-to-end TTFT. We will make this implementation detail explicit in the revision. Note that as you mention, it may be possible to do even better by integrating such calculations into the FA kernel itself, but is likely not worth the effort given the minimal overhead.
> >
> >
> > We hope these revisions make the implementation choices and comparison with Speculative Prefill and FastKV more transparent. We will ensure all of these points are reflected in the camera-ready version.

---

### Official Review · Reviewer_iFDh · 2025-11-01

**Soundness:** 3
**Presentation:** 3
**Contribution:** 2
**Rating:** 2
**Confidence:** 4

**Summary:**

This paper introduces CLAA (Cross-Layer Attention Aggregation), a training-free token ranking heuristic designed to accelerate LLM prefill by selectively processing only the most important prompt tokens. The authors first propose an Answer-Informed Oracle that establishes ground-truth token importance by aggregating attention from generated answers back to the prompt. Using this oracle, they demonstrate that existing single-layer ranking methods (GemFilter, FastKV) exhibit significant instability across layers. CLAA addresses this by aggregating token importance scores across multiple consecutive layers using maximum aggregation, deferring KV cache compression for the first 4 layers to preserve foundational representations. Experiments on LongBench, Needle-in-a-Haystack, and RULER benchmarks show that CLAA achieves up to 39% TTFT reduction compared to full KV cache while maintaining accuracy closer to the oracle baseline than existing methods.

**Strengths:**

1. The paper identifies and empirically demonstrates that existing single-layer methods suffer from significant layer-wise instability (Figure 2), which is a common limitation that impacts their effectiveness. The Answer-Informed Oracle is an important contribution that enables principled comparison of token ranking heuristics independently from architectural details. This could become a valuable tool for future research.

2. CLAA's design is straightforward - maximum aggregation across layers with delayed compression - making it easy to understand, implement, and integrate into existing systems.

3. CLAA demonstrates consistent improvements over baselines across multiple benchmarks (LongBench, Needle-in-a-Haystack, RULER) and model scales, with particularly strong performance on retrieval and multi-hop reasoning tasks. Achieving 39% TTFT reduction while maintaining accuracy close to the oracle represents meaningful benefits.

**Weaknesses:**

1. The paper is missing critical comparisons with several recent and relevant methods. Specifically, the following papers also focus on improving TTFT but are not shown in the baselines:
* Random-LTD: Random and Layerwise Token Dropping Brings Efficient Training for Large-scale Transformers
* Compressing Context to Enhance Inference Efficiency of Large Language Models
* LazyLLM: Dynamic Token Pruning for Efficient Long Context LLM Inference

2. The core technical contribution (max aggregation across layers) is relatively incremental. While effective, it's a straightforward engineering solution rather than a fundamental algorithmic innovation. The paper would benefit from deeper theoretical analysis of why cross-layer aggregation stabilizes rankings.

3. Missing real-world deployment analysis, including:
* No analysis of memory bandwidth implications when accessing KV caches across multiple layers for scoring
* No comparison of actual wall-clock time including the overhead of cross-layer aggregation computation
* Missing analysis of compatibility with optimized attention implementations (FlashAttention, which is critical for deployment)

**Questions:**

1. Can the authors provide comparisons with the baselines mentioned in the weaknesses? These are important recent methods that should be included to properly establish state-of-the-art performance.

2. How does CLAA interact with FlashAttention and other optimized attention implementations? Does the need to access attention scores across multiple layers prevent using these optimizations, potentially negating efficiency gains?

3. While ablations are provided, the method has several hyperparameters (n=4 layers, m=4 defer layers, W=8 window, kernel=7 pooling) that may need tuning for different models or tasks. Can you give the ablation of the parameters, and also analyze the sensitivity of performance from them?

4. Can you provide theoretical analysis or intuition for why maximum aggregation across layers specifically stabilizes rankings? What are the properties of attention distributions across layers that make this work?

---

> ### Author Response · Authors · 2025-11-21
>
> Thank you for the careful review and for highlighting both the strengths of the Answer-Informed Oracle and CLAA, as well as the missing comparisons and real-world deployment concerns. We address each of your points below.
>
> **Missing Prior Work**
>
> We agree that these methods are important and attempted to include them where feasible.
>
> For Random-LTD and LazyLLM, we were unable to find official, end-to-end implementations, making it difficult to compare directly against.
>
> For "Compressing Context to Enhance Inference Efficiency of Large Language Models" (Selective Context), we started from their public codebase, ported it into our LongBench evaluation harness, and ran it with LLaMA-3.2-1B-Instruct as the rewriter model and LLaMA-3.1-8B-Instruct as the main model, on the same HuggingFace and hardware used for all other baselines. We also evaluate against a second semantic compression approach (LLMLingua: https://arxiv.org/pdf/2310.05736) using the same approach.
>
> The table below reports end-to-end Time-to-First-Token (TTFT) and LongBench scores for three representative tasks. "Rewrite" is the time spent in the auxiliary model; "Prefill TTFT" is the prefill time of the main model on the rewritten prompt.
>
> | Task      | Method (Keep Rate %)        | Rewrite (ms) | Prefill TTFT (ms) | Total TTFT (ms) | Speedup vs Full KV ↑ | Score  |
> |----------:|------------------|-------------:|------------------:|----------------:|----------------------:|-------:|
> | Qasper    | Full KV (100%)   |          0.0 |            1084.3 |          1084.3 |                 1.00x | 0.487  |
> | Qasper    | CLAA (20%)       |          0.0 |             755.1 |           755.1 |                 1.44x | 0.470  |
> | Qasper    | SelectiveContext (21%) |       3133.9 |             287.4 |          3421.3 |                 0.32x | 0.186  |
> | Qasper    | LLMLingua (23%)        |        901.2 |             239.6 |          1140.8 |                 0.95x | 0.129  |
> | GovReport | Full KV (100%)   |          0.0 |           13505.0 |         13505.0 |                 1.00x | 0.377  |
> | GovReport | CLAA (20%)       |          0.0 |           10055.1 |         10055.1 |                 1.34x | 0.310  |
> | GovReport | SelectiveContext (20%)|       5642.0 |            9749.2 |         15391.2 |                 0.88x | 0.327  |
> | GovReport | LLMLingua (18%)      |       1815.0 |            8073.2 |          9888.2 |                 1.37x | 0.236  |
> | TriviaQA  | Full KV (100%)   |          0.0 |            1768.2 |          1768.2 |                 1.00x | 0.944  |
> | TriviaQA  | CLAA (20%)       |          0.0 |            1217.4 |          1217.4 |                 1.45x | 0.944  |
> | TriviaQA  | SelectiveContext (20%) |       5254.3 |             772.5 |          6026.9 |                 0.29x | 0.204  |
> | TriviaQA  | LLMLingua  (17%)      |       1682.3 |             730.0 |          2412.3 |                 0.73x | 0.470  |
>
>
> The autoregressive prompt compression step (rewrite) dominates the end-to-end latency of both Selective Context and LLMLingua. As a result, Selective Context is slower than the Full-KV baseline on all three tasks in total TTFT, and LLMLingua is at best moderately faster while also degrading quality relative to Full KV and CLAA. Semantic compression approaches could see significant benefit in amortized settings (where the same rewritten prompt can be reused many times). We will include this table and clarify this distinction in the revised version.
>
>
> **Real-world deployment**
>
> In our implementation, all methods in the paper are already evaluated using Hugging Face’s FlashAttention-2 implementation. We adopt the same design pattern as the public FastKV implementation: we subclass the FlashAttention-2 attention module to intercept the attention forward pass to compute layer-wise importance scores $S^{(l)}$ (Eq. 5) using the subset of query vectors corresponding to the observation window, $Q_{\mathcal{W}}$, against the full key matrix $K_{\text{prompt}}$. The measured TTFT in Figures 4–5 is end-to-end wall-clock time on the FlashAttention-2 stack, already including this scoring and aggregation overhead. This $Q_{\mathcal{W}}$ by $K_{\text{prompt}}$ computation represents less than 2% overhead of the entire end-to-end TTFT. We will make this implementation detail explicit in the revision.

---

> ### Author Response · Authors · 2025-11-21
>
> **Hyperparameter sensitivity**
>
> We agree that robustness to hyperparameter choices is important. For all ablation studies in the paper, we use the full LongBench dataset for each setting. We use Llama 3.1-8B-Instruct for all ablation studies. Many of the requested analyses are already in the current draft:
>
> * **Pruning layer index `l_p`:** Figure 6 (left) shows a sweep over `{3, 7, 11, 15}` for all heuristics at fixed keep rate.
> * **Cross-layer aggregation window `n`:** Figure 6 (right) reports the effect of `n = {1, 2, 3, 4, 5}` at 10%, 20%, and 40% keep rates.
> * **First uncompressed layer `m`:** Appendix B (Figure 7) sweeps `m = {0, 1, 2, 4}` and shows that deferring compression until layer 4 improves accuracy, especially at higher keep rates, justifying `m = 4` as our default.
> * **Shared settings and pooling:** Table 4 in Appendix A summarizes all hyperparameters across methods, including the observation window `W = 8` and pooling kernel size 7. These are chosen to match FastKV and to ensure that our comparisons isolate the effect of cross-layer aggregation rather than low-level tuning. In small-scale tests, varying the pooling kernel around 7 had only a minor impact on the aggregated scores.
>
> In the revision, we will explicitly point the reader to these figures and tables when discussing design choices, so that the sensitivity analysis is more visible at first read.
>
>
> **Stability of CLAA**
>
> We agree that CLAA is algorithmically simple, and we do not claim a complex new architecture. The contributions are instead:
>
> 1. The Answer-Informed Oracle, which provides a ground-truth ranking and cleanly separates ranking quality from architectural details.
> 2. The empirical finding, visible in Figure 2 and detailed per-task in Figure 8 (appendix), shows that single-layer heuristics exhibit sharp layer-wise volatility, especially in early and mid layers.
> 3. CLAA as a direct response: aggregate scores across a short window of layers to suppress that volatility while preserving simplicity and training-free deployment.
>
> We hope that the additional baselines and deployment details resolve your concerns regarding state-of-the-art comparison and practicality. Ultimately, we view the Answer-Informed Oracle as the primary contribution of this work, as it drives a more rigorous ground truth of token importance and shows where the ceiling is for this class of attention-based token pruning methods.

---

### Author Response · Authors · 2025-12-02
**Rebuttal Summary for Area Chair**

To assist the new AC, we summarize how we addressed the primary concerns driving the initial reviewer scores:

**Addressed Missing Baselines (Reviewer iFDh, Rating: 2)**

We implemented the requested baseline (*Selective Context*) and added *LLMLingua* to ensure a robust comparison against semantic compression techniques. Results in our response to reviewer iFDh show CLAA outperforms both:
*    CLAA achieves a 1.34x-1.45x speedup over FullKV while closely matching FullKV accuracy.
*    The requested baselines are either slower than the FullKV baseline (due to rewrite overhead) or significantly degrade task quality.

Since this was the primary justification for the rating of 2, we believe these favorable results directly resolve the main grounds for that assessment.

**Verified Deployment Feasibility (Reviewer iFDh, Rating: 2 and Reviewer Ke2f, Rating: 4)**

We clarified that CLAA and all other evaluated approaches already use FlashAttention-2 in the submitted paper. We achieve this by subclassing the standard attention module to compute importance scores using only the query vectors within the observation window against the key cache (similar to FastKV). This scoring/aggregation needed for CLAA accounts for < 2% of the runtime and is already included in reported end-to-end wall-clock times. Additionally, no modifications to the underlying CUDA kernels are required.

**Hyperparameter Robustness (Reviewer iFDH, Rating: 2)**

We directed reviewer Ke2f to our existing ablation studies (Fig. 6, Fig. 7, Appendix A) which already cover all CLAA hyperparameters. These ablation studies were carried out on the full LongBench dataset.

**Fairness of Speculative Prefill Comparison (Reviewer Ke2f, Rating: 4)**

We addressed concerns regarding the baseline implementation by clarifying that Speculative Prefill used the same inference stack as CLAA and other baselines. This ensures a direct, fair comparison of the token pruning heuristics on the same inference engine (HF transformers).

**Strong Support for Core Oracle Contribution (Reviewer a9cR, Rating: 8)**

We highlight that reviewer a9cR identifies the **Answer-Informed Oracles** as a “clever contribution” that effectively solves the token-ranking evaluation problem. This is the key contribution of this work.

We believe the new data provided during the rebuttal period fully resolves the specific blockers for the lower-scoring reviewers. We remain available for any further clarifications.

---

### Meta-Review · Area_Chair_xKro · 2026-01-07

**Summary:**

This paper introduces CLAA (Cross-Layer Attention Aggregation), a training-free token ranking heuristic designed to accelerate LLM prefill by selectively processing only the most important prompt tokens. Experiments on LongBench, Needle-in-a-Haystack, and RULER benchmarks show that CLAA achieves up to 39% TTFT reduction compared to full KV cache while maintaining accuracy closer to the oracle baseline than existing methods.

On the other hand, the reviewers concern about the technical contribution is relatively incremental while prior works have already shown similar ideas using attention as main metrics for token estimation (novelty), and the experiments only show limited different scale of models, plus that the ablation study actually shows the robustness to different hyperparameters are not strong enough (effectiveness), and the availability to different cases such as multi-turn conversations, real-world deployment analysis (completeness).

**Reviewer Concerns:**

The authors provide some answers to the above concerns, but the main part still seems to be outstanding. Moreover, most of the rebuttal are based on the saying "we will revise this in the new version" while there is no new version, this makes it hard to see whether the authors could address those questions.

**Reviewer Scores:**

The original scores are 2(iFDh), 4(Ke2f), and 8(a9cR). Due to the above reasons, I would not expect there will be any changes after the rebuttal discussion.

---

### Decision · Program_Chairs · 2026-01-26

Reject